# The Effect of Bystander Features on Displaced Aggression in Provocative Situations among Male Juvenile Delinquents

**DOI:** 10.3390/bs14060496

**Published:** 2024-06-13

**Authors:** Shuang Lin, Gonglu Cheng, Shinan Sun, Mengmeng Feng, Xuejun Bai

**Affiliations:** 1Key Research Base of Humanities and Social Sciences of the Ministry of Education, Academy of Psychology and Behavior, Tianjin Normal University, Tianjin 300074, China; lins000@126.com (S.L.); chgonglu@126.com (G.C.); 2100340003@stu.tjnu.edu.cn (M.F.); 2Faculty of Psychology, Tianjin Normal University, Tianjin 300387, China

**Keywords:** displaced aggression, provocative situation, bystander features, trigger, juvenile delinquents

## Abstract

Two studies were conducted to explore the influence of bystander features of displaced aggression in provocative situations among male juvenile delinquents. Study 1 examined the differences in displaced aggression between provoked male juvenile delinquents in the presence or absence of bystanders. The results revealed that provoked male juvenile delinquents exhibited significantly higher levels of displaced aggression when bystanders were present compared to when they were not. Study 2 further manipulated the bystanders’ trigger level and investigated the differences in displaced aggression exhibited by provoked male juvenile delinquents towards highly versus lowly triggered bystanders. The results indicated that after low provocation, male juvenile delinquents exhibited significantly higher levels of displaced aggression towards highly triggered bystanders compared to lowly triggered bystanders. These findings demonstrated that male juvenile delinquents exhibited a high level of displaced aggression towards bystanders in provocative situations, particularly with highly triggered bystanders. This study supported the personality and social model of displaced aggression, emphasizing that bystanders, especially those with high triggers, were more likely to become targets of displaced aggression. The current study provides references for subsequent criminal rehabilitation and crime prevention.

## 1. Introduction

Aggression refers to any behavior directed toward another individual that is carried out with the proximate (immediate) intent to cause harm [1]. Depending on the reasons and intentions behind aggression, it can be classified into reactive aggression and proactive aggression. Proactive aggression refers to planned, intentional, and voluntary aggression, while reactive aggression refers to uncontrolled aggression that individuals exhibit in response to stimuli or provocation [2]. Reactive aggression includes direct retaliatory reactive aggression and displaced aggression. Direct retaliation refers to the behavior in which individuals retaliate directly against the provocateur. Displaced aggression, as a specific type of reactive aggression, differs from direct retaliatory reactive aggression in that it involves individuals refraining from directly retaliating against the provocateur after experiencing frustration or provocation and targeting innocent individuals [3]. Displaced aggression involves two stages: being provoked and retaliating against innocent individuals. Specifically, if the provocateur is strong or powerful, people may be reluctant to directly confront the provocateur and may instead displace their aggression toward innocent targets, sometimes called scapegoats. Displaced aggression is characterized by its high unpredictability and prevalence in various contexts, which has attracted increasing attention from researchers. Previous studies have provided ample evidence for the existence of displaced aggression and its significant effect size (d = 0.54) [4]. Moreover, the prevalence of displaced aggression in schools [5], workplaces [6], and families [7] has been confirmed by existing research. The existing research findings suggested an overall increasing-then-decreasing trend in displaced aggression, reaching its peak during adolescence [8]. A survey’s result showed that the amounts and severity of juvenile delinquents’ aggression were higher than that among common adolescents of the same age [9]. Therefore, we need to identify the risk factors contributing to displaced aggression in adolescents, specifically juvenile delinquents with high levels of violence. However, adolescents do not exhibit displaced aggression towards everyone after provocation, and theoretical models and empirical research consistently indicate that bystanders who witness individuals being provoked might become targets of displaced aggression.

### 1.1. Bystander as a Target of Displaced Aggression in Provocative Situations

Provocation refers to the act of continuously provoking and instigating individuals [10]. The provocation in displaced aggression has special features, where the victim cannot directly retaliate to the provocateur due to the difference in identity or has no time to retaliate to the provocateur, eventually leading to the transfer of the attack. The personality and social model of displaced aggression emphasized that provocation at Time 1 influenced the occurrence of displaced aggression at Time 2 through individual features (such as personality traits) and target attributes (such as the innocent person’s features) [11]. For adolescents, bystanders in provocative situations are innocent people who may be more likely to become the target of displaced aggression than non-bystanders.

Although there is limited research directly examining the influence of bystanders on individuals’ displaced aggression, some related findings supported the significant role of bystanders in displaced aggression. For example, Datta et al. (2016) suggested that bystanders might have impacts on the aggression of adolescents when they are provoked [12]. Moreover, Delhove et al. (2019) examined whether participants subjected to torment by cold water would attack others in the same way [13]. The results indicated that the time participants immersed their hands in cold water positively predicted the time they wanted others to experience the same, even though researchers told subjects that the target was an innocent person. Slotter et al. (2020) explored whether individuals would attack an intimate partner or an innocent bystander after provocation by an intimate partner [14]. Specifically, participants were asked to play a game with their intimate partners and an innocent bystander, where they could earn points by playing the game or deduct points from the other two individuals. Additionally, participants could also have points deducted by the other two individuals. The results showed that participants with high levels of attachment anxiety exhibited significantly higher levels of aggression toward the innocent bystander after points were deducted by the intimate partner, compared to participants with low levels of attachment anxiety. These studies suggested that in scenarios where innocent bystanders were present during provocation, subjects might potentially engage in aggression towards these innocent bystanders. However, a previous study did not compare the differences in displaced aggression between bystanders and non-bystanders. Based on the personality and social model of displaced aggression, as well as high levels of displaced aggression among male juvenile delinquents, the current study proposed hypothesis H1: After provocation, male juvenile delinquents exhibit higher levels of displaced aggression when bystanders are present compared to when there are no bystanders present.

### 1.2. Bystander’s Trigger Further Exacerbate Displaced Aggression in Provocative Situations

Furthermore, the personality and social model of displaced aggression proposed that the trigger was the second source of displaced aggression, as provoked individuals were more likely to interpret the ambiguous behaviors of others as intentional harm towards themselves, further exacerbating individuals’ displaced aggression [11]. A trigger typically refers to individuals who are minorly offended by innocent parties, and the trigger itself does not directly lead to displaced aggression [15]. Vasquez (2009) conducted an empirical study to investigate whether cognitive load could change provoked individuals’ displaced aggression towards innocent people at different levels of trigger [16]. The study involved 80 American college students, and the results showed a significant main effect of trigger, indicating that the trigger might be the cause of the provoked individuals’ displaced aggression. Vasquez et al. (2016) suggested that a trigger as a victim feature might cause individuals to engage in displaced aggression towards innocent victims in real-life situations [17]. In other words, the occurrence of displaced aggression might be the result of the interaction between provocation and triggering. Johnson (2017) examined the roles of provocation and triggering in police officers and found an interaction between provocation and triggering [18]. Among the provoked police officers, the group with triggering showed significantly higher levels of displaced aggression compared to the group without a trigger, suggesting that a trigger might exacerbate individuals’ displaced aggression after they have been provoked. Based on the personality and social model of displaced aggression, the current study proposed hypothesis H2: Among male juvenile delinquents, provocation significantly influences the differences in displaced aggression towards different feature bystanders, with highly triggered bystanders eliciting significantly higher levels of displaced aggression compared to lowly triggered bystanders.

### 1.3. Current Study

In summary, the personality and social model of displaced aggression highlighted that provocation could directly influence displaced aggression and could also interact with individual features and bystanders’ features to affect displaced aggression. However, in addition to provocation, the features of bystanders could also impact the level of displaced aggression. Considering that puberty is a peak period for displaced aggression and that juvenile delinquents might be more likely to direct their displaced aggression towards the bystanders who triggered them [19], the current research focused on male juvenile delinquents to examine the effects of provocation on displaced aggression towards different bystanders in two studies. Study 1 investigated the impact of provocation and the presence of bystanders on displaced aggression. Study 2 further explored the effects of provocation on displaced aggression towards highly and lowly triggered bystanders to enhance the ecological validity of the research findings.

## 2. Study 1: The Effects of Bystanders’ Presence on Displaced Aggression in Provocative Situations among Male Juvenile Delinquents

### 2.1. Methods

#### 2.1.1. Participants

We estimated the required sample size using G*Power 3.1. Study 1 employed a 2 × 2 mixed design, with *f* = 0.25, *α* = 0.05, and a statistical power of 0.8; the estimated sample size was 66. Based on the principle of convenience sampling, 70 juvenile delinquents were recruited from two reform schools, located, respectively, in Guilin City and Nanning City of Guangxi Province in China, and all were male in the reform schools. Under the guidance of teachers from the reform schools, two Ph.D. psychologists recruited participants for research and received active cooperation from all participants, with none requesting to withdraw. They were randomly assigned to either the bystander group or the non-bystander group. However, two participants in the bystander group were excluded from the data due to randomly pressing buttons, resulting in a final sample of 68 participants. Among them, there were 33 participants in the bystander group (*M*_age_ = 13.61, *SD*_age_ = 1.17) and 35 participants in the non-bystander group (*M*_age_ = 13.60, *SD*_age_ = 0.77). These male juvenile delinquents were sent to reform schools for education due to illegal or criminal behaviors (such as theft, fighting, or harming others). All participants had normal or corrected vision, no color blindness, and no history of mental illness or surgical trauma. This study was approved by the ethics committee of Tianjin Normal University, and all participants signed informed consent forms and received compensation after the study. Table 1 shows the individual differences in personality traits between the two groups.

#### 2.1.2. Experimental Design

A 2 (provocation: high, low) × 2 (bystander: present, absent) mixed experimental design was used. The within-subject variable was the provocation, and the order of presentation was balanced across participants. The between-subject variable was the presence or absence of a bystander. The dependent variable was displaced aggression, measured as the proportion of choosing high-punishment options in a competitive response task. Throughout this study, all participants were explicitly informed that they were interacting with real individuals as their opponents.

#### 2.1.3. Measures

State hostile attribution bias. In this study, the State Hostility Attribution Scale (SHAS) was developed by adapting measurement tools from previous research to assess individuals’ attribution bias towards state hostility [20]. The questionnaire consisted of 10 items, and participants were asked to rate their agreement on a 7-point scale (1 = “strongly disagree”, 7 = “strongly agree”) regarding the perceived intentions of virtual opponents in a competitive response paradigm. The internal consistency reliability of this questionnaire in the present study, as measured by Cronbach’s α, was 0.84.

Trait hostile attribution bias. The WSAP-Hostility scale has been widely used to assess attribution bias towards the trait hostile attribution bias [21]. This scale included two sub-scales: attribution bias towards hostility and attribution bias towards benevolence. In this study, the sub-scale of attribution bias towards hostility was used to assess individuals’ trait hostility attribution bias. Participants were required to rate the relevance of hostility-related adjectives to ambiguous scenarios after reading 16 sentences with varying degrees of provocation. Ratings were made on a 6-point scale (1 = “completely unrelated”, 6 = “highly related”). The internal consistency reliability of this sub-scale in the present study, as measured by Cronbach’s α, was 0.82.

Displaced aggression. The Displaced Aggression Questionnaire was developed to measure displaced aggression [3]. The questionnaire included three sub-scales, totaling 31 items: anger immersion (10 items), revenge planning (11 items), and displaced aggression (10 items). Anger immersion represents the emotional component, indicating the tendency for individuals to immerse themselves in their own anger after provocation. Revenge planning represents the cognitive component, indicating the tendency to harbor resentment and plan revenge against provokers. Displaced aggression represents the behavioral component, indicating aggression towards individuals other than the original provoker. Participants rated their agreement on a 7-point Likert scale (1 = “strongly disagree”, 7 = “strongly agree”), with higher scores indicating a higher tendency for displaced aggression. The internal consistency reliability of this questionnaire in the present study, as measured by Cronbach’s α, was 0.88.

Trait self-control. The Self-Control Scale developed by Tangney et al. (2004) was used in this study [22]. It consists of 19 items and includes five dimensions: impulse control, healthy habits, resistance to temptation, work focus, and moderation of entertainment. Participants rated their agreement on a 5-point scale (1 = “strongly disagree”, 5 = “strongly agree”), with higher scores indicating stronger self-control. The internal consistency reliability of this scale in the present study, as measured by Cronbach’s α, was 0.78.

#### 2.1.4. Materials and Procedure

The modified competitive reaction time paradigm [23,24] was used to test individuals’ displaced aggression. This paradigm consisted of two stages: a passive stage (Stage 1) and an active stage (Stage 2). In the passive stage, participants were provoked by an opponent but were not allowed to retaliate. Specifically, in the passive stage, participants were instructed to compete against opponent A within a specified time. Participants saw the portrait of opponent A on the first screen of the passive stage and knew that they would compete against opponent A later in this stage. When a white dot appeared in the middle of the screen, both players had to press the space bar as quickly as possible, and the faster player won. However, in the passive stage, participants received punishment from opponent A when they lost, but they could not punish opponent A, even if they won. The provocation from opponent A in the passive stage has two levels: high provocation and low provocation. The order of presenting these levels was counterbalanced across participants. Under high-provocation conditions, opponent A chose high punishment in 80% of the trials and low punishment in 20% of the trials. Under the low-provocation condition, opponent A chose low punishment in 80% of the trials and high punishment in 20% of the trials [25]. Additionally, the experimenter emphasized the presence of bystander C throughout the passive stage for the bystander group, while the non-bystander group did not receive this instruction.

Before entering into the active stage (Stage 2), participants were explicitly informed that they would not continue competing against the same opponent A. The experimental procedure is shown in Figure 1. In the active stage, participants were ensured that they would compete against another opponent, with observer C serving as the opponent for the bystander condition during the passive stage, while a new player B served as the opponent for the non-bystander group. Compared to the passive stage, participants in the active stage did not receive any punishment when they won and could choose the intensity of punishment for the opponent based on their selected level. However, they could not punish the opponent when they lost the game. In order to exclude the influence of gender, race, age, and other factors, the specific information of the opponent was not shown during the study, but the identity of the opponent was explained as being the same as that of the participants.

The procedure is shown in Figure 2. In the post-study interview, participants were asked two questions: (1) whether they believed they faced different opponents in the passive and active stages, and (2) the reasons for their choice of punishment level in the active stage. The interview results indicated that all participants believed that the two phases of the study involved different individuals, and none of the participants refrained from attacking the innocent opponent in the active stage.

Based on relevant literature using the competitive reaction time paradigm and the threshold of physiological harm caused by white noise [26], this study used white noise as the form of punishment. Twelve different intensity levels of white noise patterns (ranging from 55 dB to 110 dB) were created using the “Cool Edit 2.1” software. Before the study, participants listened to the white noise at each decibel level through headphones and rated their personal tolerance for each noise pattern on a 5-point Likert scale (0: bearable; 1: slightly difficult to bear; 2: quite difficult to bear; 3: very difficult to bear; 4: extremely difficult to bear). In the passive stage, the noise rated as “extremely difficult to bear” (105 dB) was used as the high-intensity punishment option, while the noise rated as “slightly difficult to bear” (70 dB) was used as the low-intensity punishment option.

The procedure was as follows: (1) participants completed the task in the passive stage to activate provocation; (2) the manipulation of state hostility attribution bias was tested; (3) participants completed the task in the active stage to measure displaced aggression.

### 2.2. Results

#### 2.2.1. Manipulation Check of Provocation

The results of a repeated measures analysis of variance using state hostile attribution bias after provocation as the dependent variable showed a significant main effect of provocation, *F*(1, 66) = 31.37, *p* < 0.001, η^2^ = 0.33. The state hostile attribution bias in the high-provocation condition (*M*_high_ = 3.92, *SD*_high_ = 2.11) was significantly higher than that in the low-provocation condition (*M*_low_ = 2.24, *SD*_low_ = 1.49). Neither the main effect of the group (*F*(1, 66) = 3.50, *p* = 0.07) nor the interaction effect between provocation and group was significant (*F*(1, 66) = 0.21, *p* = 0.07).

#### 2.2.2. The Relationship between Provocation, State Hostile Attribution Bias, and Displaced Aggression

To examine the relationship between provocation and displaced aggression, coding was used for the provocation (low provocation = 0, high provocation = 1). Provocation was treated as the independent variable, state hostile attribution bias as the mediator variable, and displaced aggression as the outcome variable for the analysis of the mediation effect. The PROCESS program in SPSS 23.0 was employed, and a bootstrap test with 5000 samples was conducted. After standardizing the scores of state hostile attribution bias and displaced aggression, the results of the mediation analysis showed a significant indirect effect of hostile attribution bias (*β* = 0.19, 95% CI = [0.06, 0.35]), while the direct effect of provocation was not significant (*β* = 0.07, 95% CI = [−0.19, 0.32]). This result suggested that the participants’ aggression towards bystanders was mediated by a transfer of hostility. This finding was consistent with previous research on the model of displaced aggression [27].

#### 2.2.3. The Effects of Bystanders’ Presence on Displaced Aggression in Provocative Situations

Descriptive statistics were calculated for displaced aggression in high- and low-provocation conditions, including the mean and standard deviation. The results are presented in Table 2.

The results of a repeated measures analysis of variance using displaced aggression as the dependent variable revealed a significant main effect of provocation, *F*(1, 66) = 18.57, *p* < 0.001, η^2^ = 0.22. The proportion of aggression in the high-provocation condition (*M*_high_ = 0.45, *SD*_high_ = 0.31) was significantly higher than in the low-provocation condition (*M*_low_ = 0.36, *SD*_low_ = 0.29). There was a significant main effect of group, *F*(1, 66) = 7.81, *p* = 0.01, η^2^ = 0.11. The proportion of aggression in the presence of bystanders (*M*_yes_ = 0.50, *SD*_yes_ = 0.38) was significantly higher than in the absence of bystanders (*M*_no_ = 0.32, *SD*_no_ = 0.38). The interaction effect between provocation and group was significant, *F*(1, 66) = 4.46, *p* = 0.04, η^2^ = 0.06. For the group with bystanders, the proportion of displaced aggression in the high-provocation condition (*M*_high_ = 0.57, *SD*_high_ = 0.28) was significantly higher than in the low-provocation condition (*M*_low_ = 0.43, *SD*_low_ = 0.28), *t* = 3.41, *p* = 0.002, *d* = 0.5, as shown in Figure 3.

### 2.3. Discussion

The results of Study 1 indicated that in the presence of bystanders, male juvenile delinquents exhibited high levels of displaced aggression towards bystanders after experiencing high levels of provocation. Although previous studies did not directly examine the effects of provocation and the presence of bystanders on displaced aggression, a few studies with adolescent participants have found significant effects of provocation and opponent features on displaced aggression. For example, Reijntjes et al. (2013) studied 175 Dutch adolescents and examined how provocation and individual personality factors jointly influenced male adolescents’ displaced aggression [28]. The results showed that individuals demonstrated strong displaced aggression when they experienced strong negative feedback from opponents, and high levels of the callousness personality trait increased individuals’ displaced aggression. Based on previous studies, we proposed that after provocation, individuals might have felt embarrassed when bystanders were present. After that, individuals evoked a series of negative emotions, such as anger, shame, guilt, etc., and increased the risk of individuals engaging in high levels of displaced aggression towards bystanders.

## 3. Study 2: The Effects of a Bystander Trigger on Displaced Aggression in Provocative Situations among Male Juvenile Delinquents

The results of Study 1 indicated that in the presence of bystanders, individuals exhibited high levels of displaced aggression towards bystanders after experiencing high levels of provocation. Therefore, Study 2 further investigated the effects of bystanders’ trigger on displaced aggression in provocative situations.

### 3.1. Method

#### 3.1.1. Participants

We estimated the required sample size using G*Power 3.1. Study 2 employed a 2 × 2 mixed design, with *f* = 0.25, *α* = 0.05, and a statistical power of 0.8; the estimated sample size was 66. Based on the principle of convenience sampling, 72 juvenile delinquents were recruited from reform schools in Nanning City of Guangxi Province in China, and all were male. All participants agreed to take part in this study, no one dropped out, and they were randomly assigned to the high-provocation group (*n* = 36) or the low-provocation group (*n* = 36). All participant data were valid, and the mean age was 13.81 (*SD* = 0.49, age range 13–15 years). All participants had normal or corrected vision, no color blindness, and no history of mental illness or surgical trauma. Table 3 shows the individual differences in trait measures for the high- and low-provocation groups, but data from six participants were excluded due to extensive missing questionnaire data (questionnaire largely blank), resulting in valid questionnaire data from 66 participants.

#### 3.1.2. Experimental Design

Study 2 employed a 2 (provocation: high, low) × 2 (bystander Trigger: high, low) mixed experimental design, with provocation as the between-subject variable and trigger as the within-subject variable. The presentation order of the triggers was balanced across participants.

#### 3.1.3. Measures

Perceived threat. The measurement for perceived threat was adapted from Liu et al. (2022) [29]. It consisted of four items: “The opponent’s evaluation of me is very unfair”, “The opponent’s evaluation makes me feel embarrassed”, “The opponent makes me feel threatened”, and “The opponent has hurt my self-esteem”. Participants rated these items on a 1–7-point scale. Higher scores indicated a stronger perceived threat. In this study, the Cronbach’s α coefficient for this questionnaire was 0.81.

Other scales measuring trait hostile attribution bias, displaced aggression, and trait self-control were utilized, as in Study 1.

#### 3.1.4. Materials and Procedure

In Study 2, a manipulation of triggering was added. Specifically, in the modified two-stage (passive stage and active stage) competitive reaction time paradigm, the manipulation of high and low provocation in the passive stage remained unchanged. However, before the start of the active stage, participants were informed that during their previous game against opponent A in the previous stage, there was a bystander player (opponent C) who scored and evaluated their performance, as shown in Figure 4. The active stage involved two levels of triggering from the opponent, high trigger and low trigger, with the presentation order of high and low trigger being balanced across participants. In the high-trigger condition, opponent C would give the participant a lower score (2 out of 9) and send a negative evaluation (e.g., “You are not taking it seriously enough”). In the low-trigger condition, opponent C would give the participant a higher score (8 out of 9) and send a positive evaluation (e.g., “You are taking it seriously”) [29].

### 3.2. Results

#### 3.2.1. Manipulation Check of Trigger

The results of a repeated measures analysis of variance using perceived threat as the dependent variable showed a significant main effect of trigger, *F*(1, 70) = 23.40, *p* < 0.001, η^2^ = 0.25. The perceived threat in the high-trigger condition (*M*_high_ = 3.53, *SD*_high_ = 2.21) was significantly higher than in the low-trigger condition (*M*_low_ = 2.10, *SD*_low_ = 1.73). Neither the main effect of the group (*F*(1, 70) = 1.56, *p* = 0.22) nor the interaction effect between provocation and group was significant (*F*(1, 70) = 1.16, *p* = 0.29).

#### 3.2.2. The Effects of Bystanders’ Trigger on Displaced Aggression in Provocative Situations

Descriptive statistics were calculated for displaced aggression in the provocation conditions and triggered conditions, including the mean and standard deviation. The results are presented in Table 4.

The results of a repeated measures analysis of variance using displaced aggression as the dependent variable revealed a significant main effect of provocation, *F*(1, 70) = 5.18, *p* = 0.03, η^2^ = 0.07. The proportion of aggression in the high-provocation condition (*M*_high_ = 0.59, *SD*_high_ = 0.32) was significantly higher than in the low-provocation condition (*M*_low_ = 0.45, *SD*_low_ = 0.30). There was a significant main effect of trigger, *F*(1, 70) = 16.51, *p* < 0.001, η^2^ = 0.19. The proportion of aggression in the high-trigger condition (*M*_high_ = 0.58, *SD*_high_ = 0.31) was significantly higher than in the low-trigger condition (*M*_low_ = 0.46, *SD*_low_ = 0.32). The interaction effect between provocation and trigger was significant, *F*(1, 70) = 8.66, *p* = 0.004, η^2^ = 0.11. For the low-provocation condition, the proportion of aggression in the high-trigger condition was significantly higher than in the low-trigger condition, *F*(1, 70) = 25.24, *p* < 0.001, η^2^ = 0.27, as shown in Figure 5.

### 3.3. Discussion

The results of Study 2 also confirmed the findings of Study 1 regarding the impact of triggered conditions on displaced aggression. After low provocation, male juvenile delinquents exhibited significantly higher levels of displaced aggression towards high-trigger bystanders compared to low-trigger bystanders. Although previous research has shown an overall trend of increasing and then decreasing displaced aggression, peaking during adolescence [30], previous studies did not directly examine the effects of provocation and trigger on displaced aggression in adolescents facing different triggering bystanders. However, existing research on provocation and trigger could partially support the conclusions of this study. For example, Pedersen et al. (2008) investigated the effects of provocation by the provocateur and trigger by the innocent person on displaced aggression in adults, as well as the moderating role of target features [15]. The results showed significant main effects of provocation and trigger on displaced aggression, but the interaction between the two was not significant. For male juvenile delinquents, due to changes in their physiology and cognition, they might be more sensitive to peer evaluations compared to general adolescents, making male juvenile delinquents particularly vulnerable to experiences of shame and denigration [31]. Being observed by bystanders in provocative situations might lead male juvenile delinquents to feel that their social status was threatened, increasing the likelihood of them intending to harm others and attempting to re-establish their social status through displaced aggression.

## 4. General Discussion

This study conducted two studies to examine the impact of bystander features on displaced aggression in provocative situations among male juvenile delinquents. The findings revealed that male juvenile delinquents exhibited higher levels of displaced aggression towards bystanders in provocative situations than non-bystanders, and high-trigger bystanders might further exacerbate such behaviors. These results supported the personality and social model of displaced aggression.

### 4.1. The Effects of Bystanders’ Presence on Displaced Aggression in Provocative Situations among Male Juvenile Delinquents

The results of Study 1 indicated that after being highly provoked, juvenile delinquents exhibited intensified displaced aggression towards innocent bystanders, supporting the personality and social model of displaced aggression. This model suggested that provocation was a crucial prerequisite for the occurrence of displaced aggression, as it negatively affects individuals and subsequently exhibited aggression towards seemingly innocent targets (e.g., bystanders). Specifically, individuals found themselves in a provocative situation where they were unable to retaliate or were restricted from targeting the provocateur (e.g., power imbalance or the provocateur has left), they might redirect their aggression towards a more accessible and non-threatening target (e.g., bystander) as a scapegoat for their provocative state. Moreover, the stronger the provocation, the stronger the displaced aggression directed towards innocent bystanders.

Moreover, previous studies have found that provocation not only elicited displaced aggression in adults [32] but might also elicit such behaviors in adolescents [33]. Most research has reported the widespread occurrence of displaced aggression in various contexts for adults (e.g., workplace, family), but to some extent, it overlooked the potential threat posed by the occurrence and development of displaced aggression in adolescents. In contrast to adults, adolescents might experience a sense of humiliation following an episode of provocation, but due to their inability to retaliate against the instigator (e.g., due to unequal power dynamics or the departure of the provocateur), they might redirect their aggression towards bystanders [34]. Moreover, when bystanders appear, individuals might interpret the present of bystanders as silently condoning the aggressor’s actions. Such a perception could evoke feelings of resentment towards bystanders and may prompt them to become subjects of displaced aggression.

### 4.2. The Bystanders’ Features Influence Displaced Aggression among Male Juvenile Delinquents

Furthermore, the trigger displayed by innocent bystanders might influence the intensity of displaced aggression. Male juvenile delinquents exhibited significantly higher levels of displaced aggression towards high-trigger bystanders compared to low-trigger bystanders. Although previous research did not directly examine the impact of trigger from bystanders on displaced aggression in male juvenile delinquents, relevant findings on the trigger of innocent individuals from previous studies might provide support for the results of this study. For example, Vasquez et al. (2013) found that negative ratings and evaluations from innocent individuals significantly influenced displaced aggression as the main trigger [35]. Although these negative ratings and evaluations might not be inherently harmful, individuals who have previously experienced provocative situations might exhibit higher levels of displaced aggression when triggered by innocent individuals.

To some extent, the results of this study also contributed to the development of the personality and social model of displaced aggression. This model emphasized that provocation at Time 1 influenced the occurrence of displaced aggression at Time 2 through individual risk or protective factors. The results of this study supported the impact of provocation on displaced aggression, as proposed by the model, and also revealed that the trigger of bystanders can effectively enhance displaced aggression. After provocation, although bystanders’s triggers might seem relatively harmless, individuals served to further activate the network of hostility. As individuals might not have the time or ability to retaliate directly against the provocateur, they redirected their anger towards highly triggering bystanders.

This study has certain practical implications. The findings highlighted the importance of not only recognizing the role of provocateurs in inciting individual displaced aggression but also acknowledging the cumulative impact of bystanders’ trigger and provocative situations. In practical terms, it is essential to educate not only provocateurs to diminish harmful behaviors towards others but also bystanders in providing support to the provoked individuals or intervening to prevent further harm. Furthermore, it is crucial to equip the provoked individuals with strategies to handle provocation from provocateurs and triggers from bystanders, ultimately reducing their own inclination towards displaced aggression.

### 4.3. Limitations and Further Directions

Firstly, to facilitate the observation of displaced aggression behavior, the current study focused on male juvenile delinquents as subjects, considering that male juvenile delinquents showed a greater tendency for displaced aggression compared to common adolescents [36]. However, previous studies have also explored displaced aggression among common adolescents. For example, Lawrence et al. (2023) found that adolescents who were bullied at school reported more cases of aggression towards siblings [37]. Therefore, future research could further compare the impact of bystander characteristics in provocative situations on male juvenile delinquents and common adolescents in terms of displaced aggression.

Secondly, although previous studies have not directly examined gender differences in the impact of bystander characteristics on displaced aggression, researchers have explored gender differences in cyber victimization and cyberbullying [38]. This study found that among adolescents and adults who have experienced cyberbullying, males are more likely to exhibit cyberbullying towards innocent people than females. Therefore, based on these findings, it could be inferred that male juvenile delinquents might be more likely to engage in displaced aggression toward bystanders than female juvenile delinquents. As a result, our follow-up research would delve deeper into investigating the impact of bystander characteristics on displaced aggression and its gender differences within male and female juvenile delinquents in provocative situations.

Thirdly, the current study primarily examined the impact of verbal triggers on displaced aggression in male juvenile delinquents. Other types of triggers beyond verbal stimuli have been identified as risk factors in displaced aggression. For instance, individuals might exhibit displaced aggression towards those they believed were exploiting or mistreating them during interpersonal interactions [39]. Future research endeavors could delve deeper into the effects of various trigger types that manifested in daily experiences and their contribution to displaced aggression.

Fourthly, the current study primarily utilized behavioral indicators to investigate individual displaced aggression and bystander characteristics in male juvenile delinquents. However, additional methods, such as electrophysiological techniques (e.g., ERP and fMRI), could be employed to explore the neurobiological mechanisms underlying these behaviors. Previous studies have explored the neural underpinnings of displaced aggression, revealing that the dorsal medial prefrontal cortex could serve as a negative predictor of displaced aggression [40]. The dorsal medial prefrontal cortex was closely linked to emotion regulation and cognitive reappraisal [41], both of which played essential roles in inhibiting displaced aggression. More specifically, the dorsal medial prefrontal cortex was proposed to be involved in the process of reappraising provocative and triggered situations to regulate individuals’ anger emotions and to alleviate the impulse for displaced aggression following provocation and triggers [42].

## 5. Conclusions

This study examined the effect of provocation on displaced aggression in male juvenile delinquents through two studies, while also supporting the personality and social model of displaced aggression. Under the conditions of this study, the following conclusions were drawn: (1) Provoked male juvenile delinquents exhibited significantly higher levels of displaced aggression when bystanders were present compared to when they were not. (2) The trigger of bystanders could exacerbate the displaced aggression among male juvenile delinquents in a minorly provocative situation.

## Figures and Tables

**Figure 1 behavsci-14-00496-f001:**
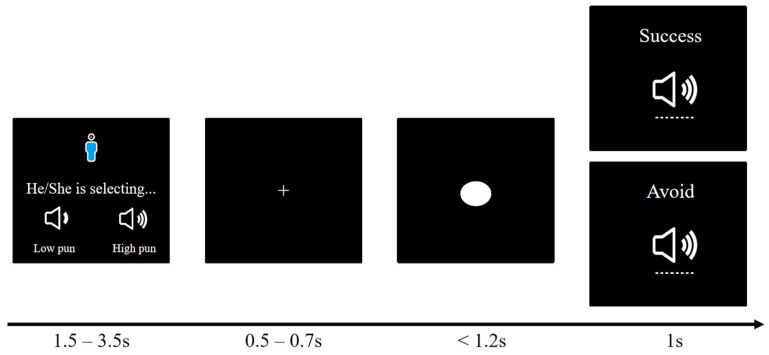
Passive stage of the modified competitive reaction time paradigm.

**Figure 2 behavsci-14-00496-f002:**
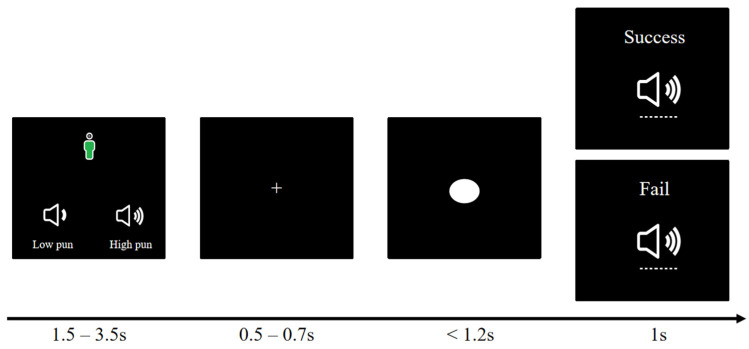
Active stage of the modified competitive reaction time paradigm.

**Figure 3 behavsci-14-00496-f003:**
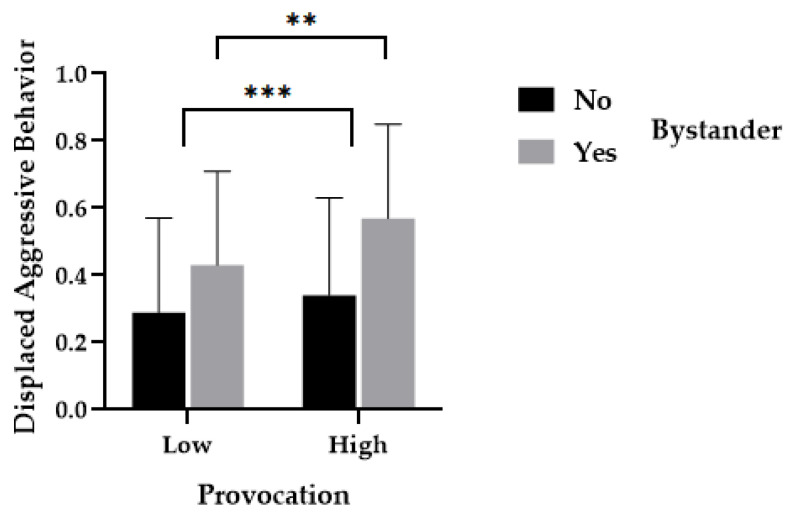
The effects of bystanders’ presence on displaced aggression in provocative situations. *Note*. ** *p* < 0.01, *** *p* < 0.001.

**Figure 4 behavsci-14-00496-f004:**
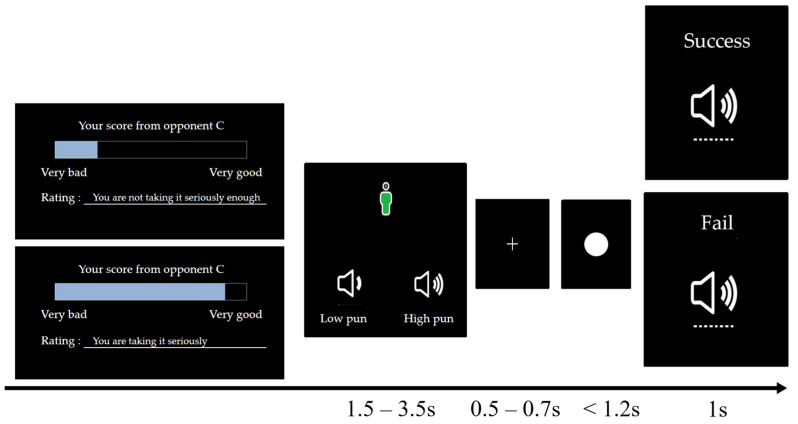
Trigger of the modified competitive reaction time paradigm.

**Figure 5 behavsci-14-00496-f005:**
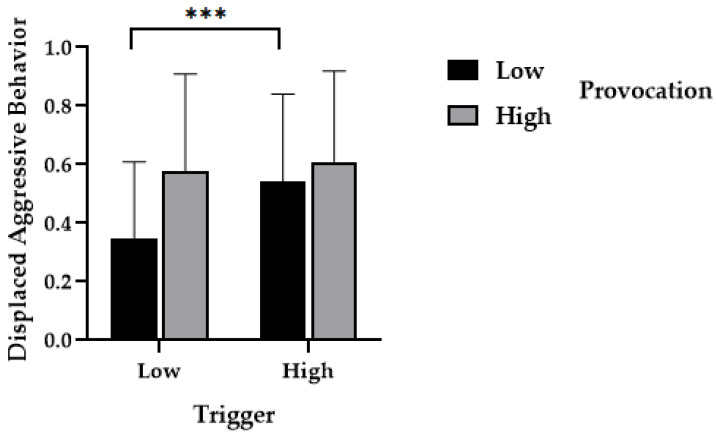
The effects of bystander’s trigger on displaced aggression in provocative situations. *Note*. *** *p* < 0.001.

**Table 1 behavsci-14-00496-t001:** Individual differences between participants in the bystander and non-bystander groups.

	Bystander Group (*n* = 33) *M* ± *SD*	Non-Bystander Group (*n* = 35) *M* ± *SD*	*t*	*p*
Trait hostile attribution bias	58.48 ± 12.33	59.17 ± 14.10	0.21	0.83
Displaced aggression	123.18 ± 37.89	121.80 ± 37.76	−0.17	0.87
Trait self-control	51.36 ± 11.83	55.77 ± 10.27	1.64	0.11

**Table 2 behavsci-14-00496-t002:** Descriptive statistics results for displaced aggression with present of bystander (*M* ± *SD*).

		Provocation	
		Low	High
Bystander	No	0.29 ± 0.28	0.34 ± 0.29
	Yes	0.43 ± 0.28	0.57 ± 0.28

**Table 3 behavsci-14-00496-t003:** Individual differences between participants in the high- and low-provocation groups.

	High-Provocation Group (*n* = 33) *M* ± *SD*	Low-Provocation Group (*n* = 33) *M* ± *SD*	*t*	*p*
Trait hostile attribution bias	59.31 ± 14.38	57.67 ± 12.25	0.59	0.56
Displaced aggression	122.39 ± 32.12	122.06 ± 37.37	0.04	0.97
Trait self-control	55.12 ± 10.26	55.00 ± 12.09	0.04	0.97

**Table 4 behavsci-14-00496-t004:** Descriptive statistics results for displaced aggression with high and low bystanders’ trigger (*M* ± *SD*).

		Provocation	
		Low	High
Trigger	Low	0.35 ± 0.26	0.58 ± 0.33
	High	0.54 ± 0.30	0.61 ± 0.31

## Data Availability

The research data will be available upon request to the corresponding author.

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
