# Peer review of "The Effect of Bystander Features on Displaced Aggression in Provocative Situations among Male Juvenile Delinquents"

_behavsci, 2024, doi:10.3390/bs14060496_

Round 1

Reviewer 1 Report

Comments and Suggestions for Authors

This is an interesting study- I do have a couple questions/comments:

- We know that deviance and impulse control are part of adolescent development. Was there a reason you did not include a control group of boys who were not adjudicated youths? This would have made your findings a lot stronger and I do think this is a weakness of your study that should be addressed.

Do you think you would see similar results for young women?

Considering your results- what would be the application for those who work with juvenile offenders and overall juveniles in general?

-

Comments on the Quality of English Language

Very minor issues in tense.  Nothing that is huge issue for me.

Author Response

Dear reviewer,

We sincerely appreciate your valuable comments and suggestions on our manuscript “The effect of bystander features on displaced aggressive behaviors in provocative situation among male juvenile delinquents ”(behavsci-3028244). Based on your comments, we have made revision to our manuscript as described below.

Point 1: We know that deviance and impulse control are part of adolescent development. Was there a reason you did not include a control group of boys who were not adjudicated youths? This would have made your findings a lot stronger and I do think this is a weakness of your study that should be addressed. Do you think you would see similar results for young women?

Response 1: Thanks for your comments. We totally agree with your point that including non-offending male adolescents as a control group could provide stronger support for our findings. We have acknowledged this point as indicated in blue font in section 4.3 on Limitations and future directions. We chose male juvenile delinquents as subjects considering their high-level displaced aggression compared to the common adolescents, which facilitated our observation of displaced aggression in current study. Although the current study did not include non-offending male adolescents as a control group, we proposed that the results still addressed the research questions: (1) when bystanders were present, male juvenile delinquents after provocation exhibited significantly higher levels of displaced aggression compared to when bystanders were absent, and (2) bystanders’ triggers exacerbated displaced aggression among male juvenile delinquents.

Point 2: Considering your results- what would be the application for those who work with juvenile offenders and overall juveniles in general?.

Response 2: Thank you for your comments. Although previous studies did not directly investigate the gender differences in the impact of bystander features in provocative situations on displaced aggression, researchers have examined gender differences in cyberbullying victimization and cyberbullying behavior [1]. The findings of this study suggested that among adolescents and adults who have experienced cyberbullying, males were more likely to exhibit cyberbullying behavior towards innocent individuals than females. Based on the above result, it could be speculated that male juvenile delinquents might exhibit higher levels of displaced aggression towards bystanders in provocative situations compared to female juvenile delinquents. We sincerely appreciate your interesting idea, and our future research will explore the impact of bystander features in provocative situations on displaced aggression and its gender differences between male and female juvenile delinquents. We have included this idea in the Limitations and future directions section, as shown in blue font in section 4.3.

  1. Zsila, Á.; Urbán, R.; Griffiths, M.D.; Demetrovics, Z. Gender differences in the association between cyberbullying victimization and perpetration: The role of anger rumination and traditional bullying experiences. Int. J. Ment. Health. AD. 2019, 17, 1252–1267.https://doi.org/10.1007/s11469-018-9893-9

Point 3: Quality of English Language: Pay attention to the correctness of the recording of the times.

Response 3: Thank you for your suggestions. We have repeatably reviewed and revised the English expressions, as indicated by the blue font in the document.

We sincerely appreciate your valuable comments and suggestions. Should our revisions be inadequate or incorrect, we kindly ask for the opportunity to further revise the manuscript please. Your feedback is invaluable, and we are grateful for your assistance and support throughout this process. Thank you once again for your guidance and support.

Reviewer 2 Report

Comments and Suggestions for Authors

First of all, 

the article deal with a very interesting topic. Despite this, changes are necessary.

In the abstract, due space must be given to discussion and conclusions. In the abstract comments on the results and practical implications are totally missing.

Keyword: pay attention, two words are divided by comma and not semicolon. Use single or double words for each keywords, not 3.

About aggressive behaviors you should explain in more detail what aggressive behaviour consists of.

In the current study you should explain which are the research questions and not only the hypothesis (pay attention, you wrote hypotheses).

It would be better to specify how the sample was recruited, how many people were contacted and what percentage agreed to participate.

Since these are two studies (talking about experiments is dangerous and misleading!) interesting discussions need to be extended and research questions need to be brought up before the relevant discussion. 

a section on limitations and future directions is missing.

The bibliography should be expanded and, above all, updated. Too many references are before 2010.

Comments on the Quality of English Language

English has some typos and language that is not always scientific and academic.

Proofreading is necessary

Author Response

Dear reviewer,

We sincerely appreciate your valuable comments and suggestions on our manuscript “The effect of bystander features on displaced aggressive behaviors in provocative situation among male juvenile delinquents ”(behavsci-3028244). Based on your comments, we have made revision to our manuscript as described below.

Point 1: In the abstract, due space must be given to discussion and conclusions. In the abstract comments on the results and practical implications are totally missing.

Response 1: Thank you for your suggestion. Your suggestion has been very helpful for our revisions. We have made modifications to the Abstract section, in which supplemented the relevant statements about the discussion, conclusion, and implications, as indicated in blue font in the abstract section. In addition, at the end of section 4.2 of the General Discussion, we have specifically added the practical implications of this study, as shown in the blue text in section 4.2 of the General Discussion.

Point 2: Keyword: pay attention, two words are divided by comma and not semicolon. Use single or double words for each keywords, not 3.

Response 2: Thank you for your advice. We have modified the format of the keywords basing on the journal requirements and highlighted them in blue font.

Point 3: Introduction: About aggressive behaviors you should explain in more detail what aggressive behaviour consists of.

Response 3: Thank you for your suggestion, Your suggestion has been very helpful for our revisions. According to your request, we have provided more details of the components of aggression in the Introduction section, highlighted in blue font.

Point 4: Introduction: In the current study you should explain which are the research questions and not only the hypothesis (pay attention, you wrote hypotheses).

Response 4: Thank you for your advice, we totally agree with your viewpoint. According to your request, we have summarized our research question in the section 1.3 of the Introduction, highlighted in blue font.

Point 5: Methods: It would be better to specify how the sample was recruited, how many people were contacted and what percentage agreed to participate.

Response 5: Thank you for your suggestion. We have provided more details of the recruitment process of participants in the Participants section of the research methods, as indicated in blue font in the Participants section.

Point 6: Discussion: Since these are two studies (talking about experiments is dangerous and misleading!) interesting discussions need to be extended and research questions need to be brought up before the relevant discussion.

Response 6: Thank you for your suggestion, we totally agree with your viewpoint. We have revised the term "experiment" in the text to "study" and have enhanced and expanded the Discussion section, as indicated in blue font in the Discussion section.

Point 7: Limitation and further direction: A section on limitations and future directions is missing.

Response 7: Thank you for your suggestion. We have added the research limitations and future directions, as shown in blue font in section 4.3.

Point 8: References: The bibliography should be expanded and, above all, updated. Too many references are before 2010.

Response 8: Thank you for your suggestion. We have reviewed the references and replaced some of the references prior to 2010, as indicated in blue font in the Introduction section, Discussion section and References section.

Point 9: English has some typos and language that is not always scientific and academic.

Response 9: Thank you for your advice. We have repeatably reviewed and revised the English expressions, as indicated by the blue font in the document.

We sincerely appreciate your valuable comments and suggestions. Should our revisions be inadequate or incorrect, we kindly ask for the opportunity to further revise the manuscript please. Your feedback is invaluable, and we are grateful for your assistance and support throughout this process. Thank you once again for your guidance and support.

Round 2

Reviewer 2 Report

Comments and Suggestions for Authors

Ok, now it's fine!